# An Enhanced Parallelisation Model for Performance Prediction of Apache Spark on a Multinode Hadoop Cluster

**Nasim Ahmed** [1,*] , **Andre L. C. Barczak** [1] , **Mohammad A. Rashid** [2] **and Teo Susnjak** [1]

1   School of Natural and Computational Sciences, Massey University, Auckland 0745, New Zealand;
    a.l.barczak@massey.ac.nz (A.L.C.B.); T.Susnjak@massey.ac.nz (T.S.)
2   Department of Mechanical and Electrical Engineering, Massey University, Auckland 0745, New Zealand;
    m.a.rashid@massey.ac.nz
*   Correspondence: nasim751@yahoo.com

**Abstract:** Big data frameworks play a vital role in storing, processing, and analysing large datasets. Apache Spark has been established as one of the most popular big data engines for its efficiency and reliability. However, one of the significant problems of the Spark system is performance prediction. Spark has more than 150 configurable parameters, and configuration of so many parameters is challenging task when determining the suitable parameters for the system. In this paper, we proposed two distinct parallelisation models for performance prediction. Our insight is that each node in a Hadoop cluster can communicate with identical nodes, and a certain function of the non-parallelisable runtime can be estimated accordingly. Both models use simple equations that allows us to predict the runtime when the size of the job and the number of executables are known. The proposed models were evaluated based on five HiBench workloads, Kmeans, PageRank, Graph (NWeight), SVM, and WordCount. The workload's empirical data were fitted with one of the two models meeting the accuracy requirements. Finally, the experimental findings show that the model can be a handy and helpful tool for scheduling and planning system deployment.

**Keywords:** big data processing; Apache Spark; execution time prediction; performance prediction; modelling

## 1. Introduction

An increasing amount of data is coming from many applications, and it has become a challenging task to store and process big data efficiently [1]. In the last decade, researchers proposed and developed efficient distributed parallel file systems, such as MapReduce [2] and Spark [3], which provide various functions, including fault-tolerant, high scalability, open access [4], and simple application programming interfaces (APIs). Spark got prompt attention from professionals and researchers because of those features and fast data processing [5]. Spark can support of a wide range of data processing libraries, such as SQL spark for structured data processing; MLlib; and GraphX for machine learning, image processing, and streaming [6]. Besides, it can also store batch and streaming data and process this data using the applications and store the results in HDFS.

Spark introduced a new data abstraction technique called resilient distributed dataset (RDDs) [3] that improves multiple applications' performances. Its application execution time is an essential factor in measuring real-time processing where the optimum execution time can be obtained based on accurate resource allocation. Spark's performance expansively depends on the suitable selection of parameters, as this system has more than 150 parameters, and the selection and configuration of these parameters are challenging. The users need to adjust the configuration parameters as per the cluster resources; else, the cluster's performance degrades significantly. Indeed, it is essential to select and configure the parameters that play an important role in system's performance [7]. In the recent past, researchers proposed number of techniques, such as trial-and-error [8], cost-based (analytical) [9], and machine

learning modelling [10,11]. However, all these techniques are either time-consuming or require large amounts of training and test data [12]. There are many issues practitioners may encounter when trying to model the performance of a cluster. One standard option to create a model is the use of machine learning algorithms, but this requires that enough sample runs are acquired, and this can take time. If not enough samples are acquired, capturing diverse data points, the accuracy of the model may suffer. Furthermore, machine learning can be a black box for the practitioner, and finding a simple model would be very useful because that would minimise the need to run workloads repeatedly too many times. Therefore, the following research question arises: *"What parallelisation model for a Hadoop cluster can be found and implemented quickly and efficiently in order to improve the performance prediction of a job?"*

Any algorithm can be parallelised, but not all algorithms can run efficiently in parallel machines such as a Hadoop cluster. It is a common phenomenon that the parallel performance depends mostly on how the algorithm operates and how nodes communicate to one another. In any parallel system, two of the most important parameters that will determine the runtime are the size of the job and the number of available executors (here executors can be interpreted as CPUs, or nodes of a cluster). Other parameters can drag the performance down, but they will not necessarily increase the performance. For example, if not enough memory is available to a job, this will increase runtime. If the minimum amount of memory is available, more memory will not make the job run faster. The number of available executors is very important, especially when the algorithm being parallelised requires communication that is not needed when the same algorithm is implemented and run on a single executor. For example, some algorithms are embarrassingly parallel (a term coined in the 90s), meaning that no extra work is needed when the job is parallelised. In this case, the speed-up is proportional to the number of processors available. In other cases, the speed-up can be super-linear, as in the case of searching algorithms running in parallel. Unfortunately, there are also groups of algorithms that do not present this optimistic speed-up [13]. One important factor that causes the degradation of performance is the fact that the algorithm may require extra communication and I/O operations that are inherently serial in nature.

The motivation for this paper was to extend our previous work, where we proposed a simple model to predict runtime as a function of number of executors [14]. The novelty of that work was the consideration of the importance of the amount of data in such performance prediction models. Accordingly, we extend the previous model and propose new parallelisation models that consider the number of executors and the the amount of data simultaneously. To the best of the authors' knowledge, such models have not been published in the literature before. These new runtime performance prediction models rely on simple equations. They can potentially be as fast and as accurate as models created using machine learning. The authors have experimentally confirmed that the proposed ideas can be very useful for the runtime performance prediction of Spark jobs on the Hadoop cluster because they require minimum training data to achieve good predictions in less time.

The key contributions of this paper are as follows:

- We introduced two distinct parallelisation models for performance prediction of Spark jobs on Hadoop cluster. Each model is based on a different communication pattern between the nodes of a Hadoop cluster.
- We accomplished extensive experimental work. The authors analysed and verified the performance pattern based on two main parameters, the number of executors and the amount of data for each job. The data reliability was verified by running each workload at least three times.
- We evaluated our models on five HiBench workloads in order to test the data fitting accuracy. Our results show that the experimental data fitted one of the models accurately, and the fitness was compared with Amdhal's law, Gustafson's law, and Ernest's model. The data fitness was compared based on two criteria, Rsquared and RRSE.

The remainder of this paper is organised as follows: Section 2 provides a brief overview of Apache Spark. Section 3 presents some interesting Spark performance prediction based on a recently published Hadoop cluster-related study. In Section 4, we discuss existing models for runtime prediction for a Hadoop cluster. In Section 5, we describe a parallel model based on a fully connected network, and discuss the motivation for this model. In Section 6, we explain the experimental setup and present the workload execution and show the DAG of stages. Section 7 presents the results and analysis; in particular, it shows how the different equations fit the data. Finally, in Section 8 we present our conclusions with a discussion on the future developments for the model.

## 2. Apache Spark Platform

Matei Zahari developed Apache Spark at UC Berkely's AMPLab in 2009 [3]. In 2010, Spark became an open-source project. Spark has since been very popular and serves as an alternative to the MapReduce model for open access, high-performance [4], and real-time data processing [15]. Spark presents a new way to process data faster, and its uses are in data analytics, big data processing, and machine learning. The major advantage of Apache Spark for machine learning is its end-to-end capabilities. As per the Datanyze market research [16], Apache Spark's market share is almost 6.40% with more than 2770 companies globally. As per enlyft data [17], 59% of customers of Apache Spark are in the United State, 6% are in the United Kingdom, and 6% are in India.

Many programming languages, such as Python, Scala, Java, and SQL APIs, are embedded within this technology for use and development purposes. Compared to Hadoop, Spark offers a hundred times faster memory and ten times faster performance on disk. Due to its memory, Spark increases the performance of the application. Spark is an ecosystem which consists of various components, such as Spark SQL, Spark Streaming, Mllib, GraphX, and core API components. These components are designed to work closely to the core, and an application can be developed based on their libraries. Apache Spark offers well-defined architecture. In this architecture, the two main abstractions are Resilient Distributed Datasets (RDDs) and Directed Acyclic Graph (DAG). Generally, in the Spark cluster, an RDD collects the data and splits the data into partitions; then, this partitioned data are stored in the memory on worker nodes and parallel operations are performed. Spark's RDD supports two types of operations, transformations and actions. A transformation uses the existing data to create a new dataset, and actions perform computations on the dataset and return their values to the driver program [18]. In Apache Spark, DAG consists of sequences of vertices and edges. Any job submitted in Spark creates a DAG and forwards the job into the stage level, where every stage is comprised of tasks based on input data and the RDD partition.

Apache Spark architecture uses master–slave systems with driver programs. The driver program runs as a master node, and the executors run as slave nodes. The executors start their processes once they receive the input file and continue until the job is completed. In this case, the executors keep themselves active the entire time and use multiple CPU threads for the task in parallel. The driver program creates the SparkContext and stores all the components. Spark driver and SparkContext look after the job execution in the cluster. In Spark, a job is executed in one or multiple physical units, and the jobs are divided into smaller sets of tasks at this stage. A single spark job can trigger many jobs that are dependent on the parent stage. Thus, the submitted job can be executed in parallel. Spark runs submitted jobs in two stages: ShuffleMapStage and ResultStages. ShuffleMapStage is an intermediate stage where the output data are stored for the following stages in the DAG. The ResultStages is the final stage of this process that applies a function to one or multiple partitions of the target RDD.

For any given work, the number of executors, the amount of data, and the number of threads play vital roles in the performance [19]. The block manager acts as a cache storage for a user's program when executors allocate memory storage for the RDDs. Spark runs on a Hadoop cluster with Apache YARN (Yet Another Resource Negotiator) [20]

as a framework for resource management and job scheduling or monitoring, in separate domains; and Apache Ambari manages, monitors, and profiles the individual workloads running the Hadoop cluster. Figure 1 shows a typical Spark cluster architecture.

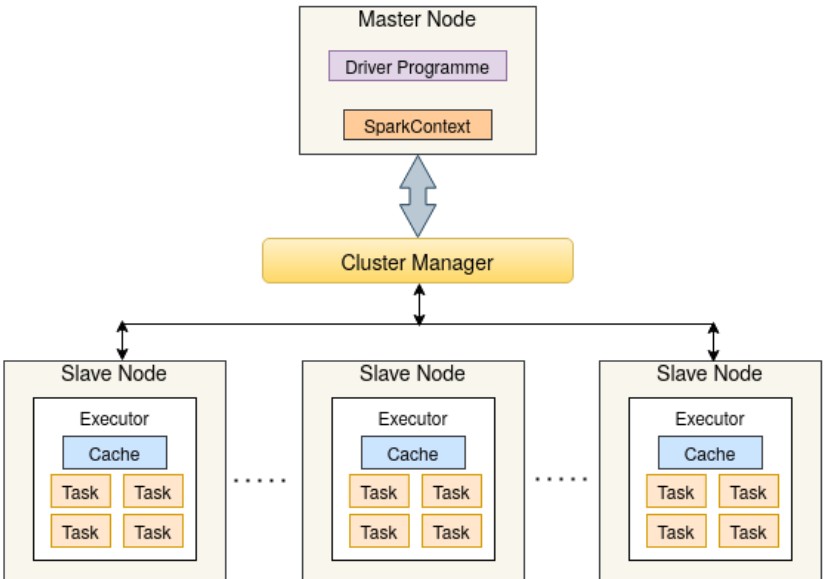

**Figure 1.** A typical Spark cluster architecture.

## 3. Related Work

The current state-of-art in Spark performance predictions of big data has received widespread attention from researchers. Researchers have proposed several exciting works based on trial-and-error [8], grey-box modelling [21], black-box modelling [12], and machine learning approaches [22,23]. In this section, we present a review of the literature published in the recent past.

Petridis et al. [8] presented a trial-and-error methodology to predict the execution time of a Spark job. This work highlighted how the number of cores and parallelism play significant roles in the performance. There were twelve parameters considered with three benchmark applications—sort-by-key, shuffling, and Kmeans. They obtained significant performance improvements by using KryoSerializer rather than using the default Java Serializer, and the speed-up achievement was 10-fold. In their second work [24], they proposed an alternative systematic methodology for parameter tuning which can be applied to any computing infrastructure. They identified that the number of cores of the Spark executor has most impact; and the level of parallelism—for example, the number of partitions per participating core—plays a significant role in maximising the performance improvement.

Muhammad Usama Javaid et al. [25] proposed a robust performance model based on a machine learning (ML) algorithm. In order to train the ML algorithm, they used various amounts of input data, sets of spark parameters, and features.

A complex data-driven workflow application was proposed by Gulino et al. [26] where they combined ML and an analytical model to predict the execution times of arbitrary complex workflow applications.

Cheng et al. [27] proposed a machine-learning-based, efficient, performance prediction model for Apache Spark. This technique was capable of predicting the execution times accurately for the given application and configurations. At the stage level, Adaboost was used to build the model. They used projective sampling and data mining techniques to mitigate the modelling overhead. They claimed that the proposed model offers three advantages: no prior assumption of configuration parameters, stand-out robustness and stability, and less overall cost for the modelling process. They found that the average prediction error of the model was only 9% as compared to the other techniques.

In their second work, Cheng [11] proposed combined multi-object optimization (MOP) and an Adaboost algorithm to find the optimal configuration of parameters and predict the model's performance. They evaluated the system with six Spark benchmarks. Five different datasets were used to analyse the performance. They claimed that the model can find the appropriate configuration setup and minimise the time and computational cost. The average improvement in computational cost was about 35% to 40%.

Aziz et al. [28] presented the resource management and data processing, the system processing time and speed-up, and the impact of persistence of resilient distributed datasets (RDDs) in Spark based on machine learning algorithms. In this analysis, the appropriate storage level of execution time was presented for Spark using a machine learning algorithm in RDD. They found that the speed-up does not improve by adding additional nodes, and the performance is degraded; and the total processing time increased significantly. There were many factors behind this degradation: among them, the most significant reason was the 100% allocation of cores to executors.

Boden [29] proposed a representative set of machine learning algorithms (supervised and unsupervised) to investigate large-scale datasets. The mathematical variation and appropriate system parameters were tuned for the amount of data and dimensionality of the data. The author reported that machine learning algorithm problems exhibit very high dimensionality due to data scaling and model size scaling. Therefore, they focused on the aspects likely affecting scaling the data and scaling the model's dimensionality. Their study found that as the the amount of data was increased, the system exhibited linearly increments in time consumed.

A cost–benefit Spark performance prediction model based on a machine learning algorithm was proposed by Maros [30]. They have proposed both black-box and grey-box models based on four machine learning algorithms. They considered three different aspects: the amount of training data, platform configurations, and workloads. They compared their model with Ernest [31]. They found that the performance estimation error was better than Ernest when the dataset extrapolation was required. Mustafa [10] proposed a new platform to predict the execution time for SQL queries and machine learning applications. This technique is very similar to the grey-box model. They applied three different approaches which used existing methods to predict the execution times of the queries. The authors claimed that the SQL query workload produced less than 10% error, whereas the machine learning workload produced less than 25%.

An exciting system was proposed by Amannejad et al. [32], which can predict the execution time in a short time. In this method, minimum resource settings are considered, which do not have complex dependencies and parallel stages. An application is used to analyse the work log files. This method requires two reference files, and the files are relatively small. This method had excellent accuracy regarding execution time, where the average prediction error of the workloads was about 4.8%. Unlike this work, they considered only a single node cluster, not a real cluster environment.

In a related but alternative model, PERIDOT was presented by Amannejad et al. in their second paper [33]. A small subset of input data and fixed limited cluster resources settings were considered to get quick execution time. They analysed the logs from both the executions and checked the internal dependencies between the internal stages. There were eight HiBench workloads used this experiment. They reported that the data partitions and the number of executors had significant impacts on execution time. This method had an overall mean prediction error of 6.6%, except for naive prediction techniques.

We summarise the different approaches in Table 1.

Unlike our approach, other approaches described in the literature may require time to modify several default parameters, which are very complex and tedious to work with. Apart from this, machine learning models usually require a large number of experiments in order to generate enough data for model training. Our proposed models need very few experiments, fitting the data obtained into simple equations. The equations can also

give some insights into the pattern of communication between the nodes when running Spark jobs.

**Table 1.** The recent approaches to Spark performance prediction.

| References | Approach/Method | System/Environments |
|---|---|---|
| Cheng et al. [27] | Machine Learning | Efficient performance prediction for Apache Spark. |
| Ahmed et al. [34] | Comprehensive Trial-and-Error | Apache Hadoop and Apache Spark for large scale datasets. |
| Al-Sayeh et al. [21] | Gray-box modelling | Runtime prediction of Spark jobs. |
| Shah et al. [33] | PERIDOT | Quick execution time predictions for Spark applications. |
| Aziz et al. [28] | Machine Learning | Resource management for efficient performance of Apache Spark. |
| Gounaris et al. [24] | Alternative Systematic | Spark parameter tuning. |
| Mustafa et el. [10] | Machine Learning | Predicting execution time of Spark jobs. |
| Chao et al. [35] | Gray-box modelling (Machine Learning) | Spark performance model for accuracy improvements. |
| Petridis et al. [8] | Trial-and-Error | Spark parameter tuning. |

## 4. Parallelisation Models

### 4.1. Amdahl's Law and Gustafson's Law

If no communication between the various executors is needed to run a job, the job is called "embarrassingly parallel" [13]. The implication of having no need to communicate between different executors is that the speed-up is proportional to the number of executors; i.e., if one executor takes time $t$, then $n$ executors will take time $\frac{t}{n}$. However, any small portion of the job that is not parallelisable can bring major consequences for parallel performance. In this case, the linear speed-up achieved by adding more executors (in the form of CPUs or cores, or separate nodes) may decline sharply.

Amdahl came up with a generic equation to predict the speed-up factor of a parallel application as a function of the number of processors [36]. The equation considers that parts of the application (or job, or workload) are inherently serial in nature and would not be parallelisable.

$$S(nexec) = \frac{nexec}{1 + (nexec - 1) f_{np}} \tag{1}$$

where $S(nexec)$ is a function that represents the speed-up as a function of the number of executors, $nexec$ is the number of executors (often interpreted and nodes or CPUs available in the infrastructure), and $f_{np}$ is the factor of non-parallelisable portions of a job. $f_{np} = 0$ represents a perfectly parallelisable job that will yield full speed-up (e.g., if there are 10 executors available, the job will run 10 times faster, or $S(10) = 10$).

From Equation (1), and considering that a single processor takes time $t$ to run a certain workload, the predicted runtime running on multiple processors would be:

$$runtime = \frac{(1 - f_{np}) t}{nexec} + f_{np} \ t \tag{2}$$

where $t$ is a hypothetical runtime needed to run a job in a single executor.

If we consider the size of the job, we can modify Equation (2) to:

$$runtime = a\, f(Size) \left( \frac{(1 - f_{np})}{nexec} + f_{np} \right) \tag{3}$$

where *a* is a constant coefficient, and $f(Size)$ is a function that reflects the growth of the runtime with increasing sizes (an approximation of the algorithm complexity). As most of the workloads implemented in HiBench are either linear or quadratic, $f(Size)$ can be replaced by either *Size* or $Size^2$.

An example of what Amdahl's law means for different serial factors and numbers of executors is shown in Figure 2. It shows the influence of both parameters on the runtime of a simulated job with a fixed dataset size.

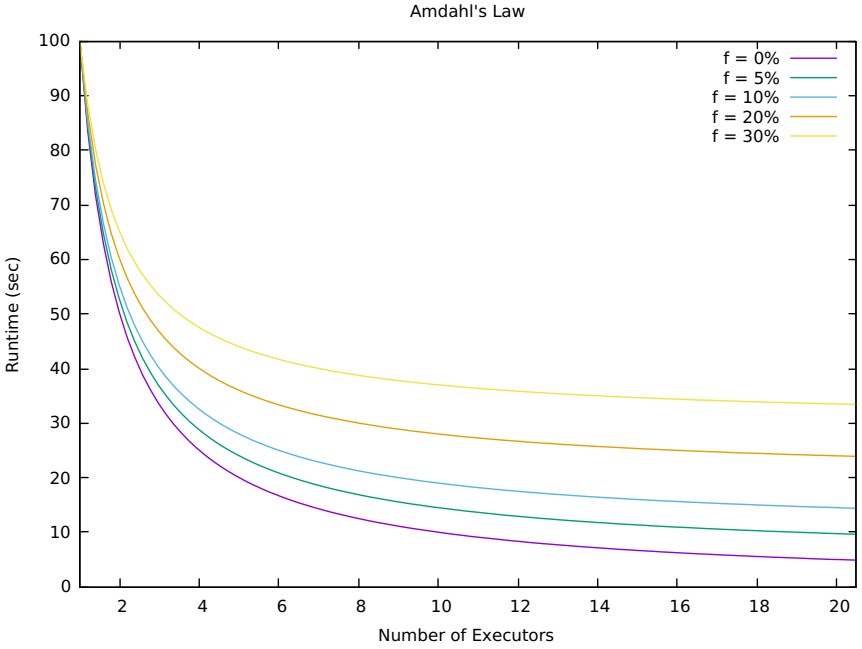

**Figure 2.** Amdahl's law for various serial factors and numbers of executors.

A few years after Amdahl's publication, Gustafson argued that the percentage of the serial part of a job is rarely fixed for different problem sizes [37]. In Amdahl's law even a small percentage of serial work can be detrimental to the potential speed-up after adding more executors. Gustafson noticed that for many practical problems the serial portion would not grow with an increase in problem size. Gustafson's speed-up equation is:

$$S(nexec) = nexec + (1 - nexec)\, f_{np} \tag{4}$$

Additionally, the runtime equation as a function of *Size* and *nexec* can be written as:

$$runtime = \frac{a\, f(Size)}{nexec + (1 - nexec)\, f_{np}} \tag{5}$$

Both Amdahl's and Gustafson's equations show that runtime will always go down as the number of executors increases. However, often in practice the communication can impose an overhead, so runtime might increase after a certain limit on the number of executors. We compare Equations (3) and (5) to our own model of parallelisation, as discussed in the next section.

An example of what Gustafson's equation (5) means for different serial factors and numbers of executors is shown in Figure 3. It shows the influences of both parameters on the runtime of a simulated job with a fixed dataset size.

### 4.2. A Model Using a 2D Plate Communication Pattern

In our previous work [14] a model using a 2D plate communication pattern based on a description by [13] was proposed and tested. In that work, we used the following equation, which is a function of *nexec* only (fixed problem sizes):

$$runtime = \frac{a}{nexec} + b \sqrt{nexec} \qquad (6)$$

where *a* and *b* are coefficients, and *nexec* is the number of executors.

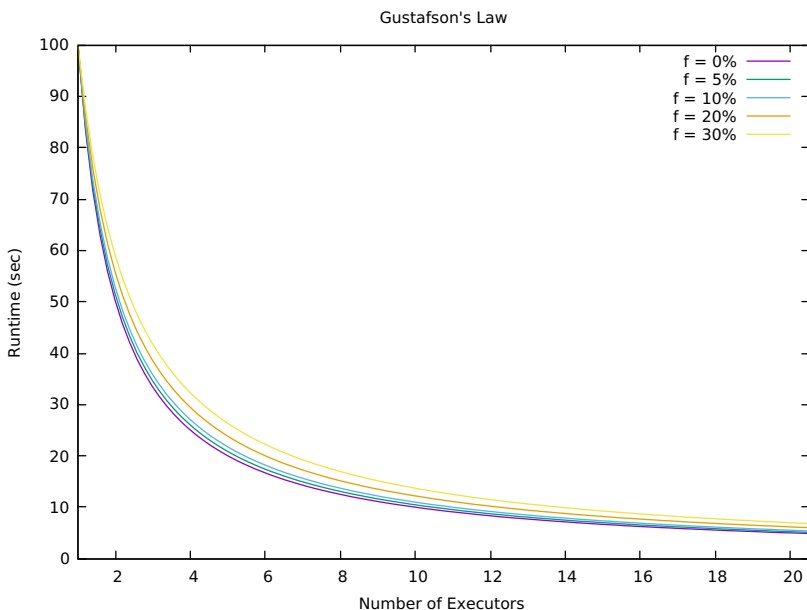

**Figure 3.** Gustafson's law for various percentages of serial work.

The equations were based on a communication model where each node has to exchange information with certain neighbours, but not all. Figure 4 shows an example of the boundaries of communication between nodes for this model.

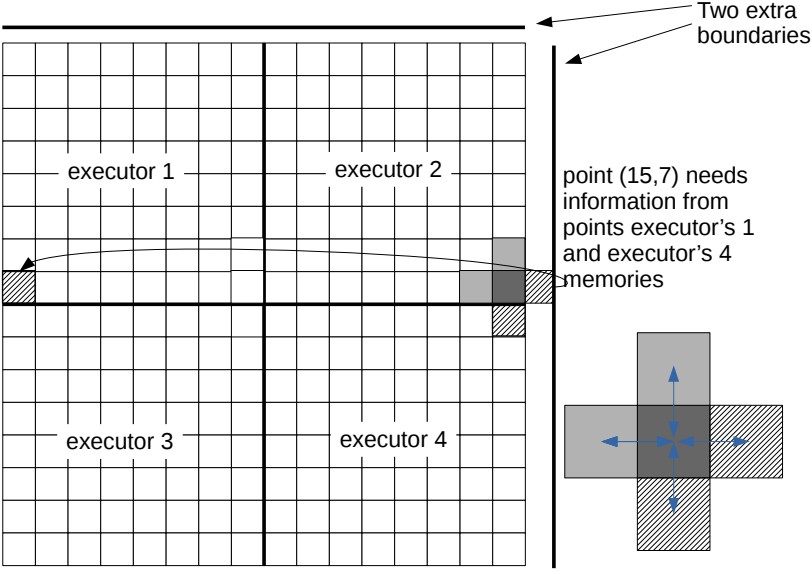

**Figure 4.** A 2D plate's homogeneous node communication.

The second part of Equation (6) is a function of $sqrt(nexec)$. We have also experimented with a different function, $nexec^c$, where $c <= 1$, and found that for some data it worked even better. Furthermore, based on [31], we added a constant term $d$ to the equation to improve the fitting. Therefore, Equation (6) can be rewritten as:

$$runtime = \frac{a\, f(Size)}{nexec} + b\, g(Size)\, nexec^c + d \qquad (7)$$

where $f(Size)$ is a function representing the time complexity of the workload when it runs on one executor, and $g(Size)$ is a function that indicates the growth of the communication and overhead when parallelising the job. After preliminary experiments, we found that $g(Size) = Size$ works well.

For linear algorithms, $f(Size) = Size$, so we can rewrite Equation (7) as:

$$runtime = \frac{a\, Size}{nexec} + b\, Size\, nexec^c + d \qquad (8)$$

If the workload is quadratic, $f(Size) = Size^2$, and Equation (7) can be rewritten as:

$$runtime = \frac{a\, Size^2}{nexec} + b\, Size\, nexec^c + d \qquad (9)$$

## 5. An Enhanced Model for Runtime Prediction

The model described in Section 4 has its limitations, as it assumes that each node would only communicate with a small number of neighbour nodes. We observed that although the model fits the data well for some workloads, it still may not reflect the communication that may be required when using HDFS, where copies of the data may be anywhere in the cluster. It is a known issue that for different algorithms, different communication patterns emerge [13], as the algorithm itself may require data located elsewhere or computations carried out by other nodes. In the proposed models, we took communication as a single factor, as this simplifies the model.

In order to expand the model to include both the number of executors and job sizes, we decided to reformulate the model. Thus, we considered that a node (where the executors have CPU resources) can communicate with any other node in the cluster. Although the communication pattern is not known for a black box implementation, we can infer what is happening through the empirical data acquired by running the same workload with many sizes and numbers of executors.

The communication pattern is assumed to be of the shape of a fully connected graph (see Figure 5). The assumption is that a function (unknown) of the size and number of executors drives the extra runtime needed to complete the communication between nodes. The extra runtime is, of course, compensated by the extra nodes involved in the job. Therefore, two components of the equation drive the runtime in opposite directions: the extra nodes will divide the processing to run the job, but communication between them requires extra time. The basic parallel equations start as:

$$runtime = \frac{t}{nexec} + t_{serial} \qquad (10)$$

where $t$ is the runtime for a job of a certain size to run in a single executor (no communications involved), $nexec$ is the number of executors, and $t_{serial}$ is the serial portion of the job that cannot be parallelised, here considered to be communication overheads and any other overheads required to run the job in parallel.

If the size is added to the model, we need to know the algorithmic complexity of the implemented code.

$$runtime = \frac{f(Size)}{nexec} + t_{serial} \qquad (11)$$

For simplicity, we assume that each node will communicate with every other node, and that the communication (be it HDFS or partial computations being exchanged between nodes) is symmetric and homogeneous. This makes the growth of $t_{serial}$ a function of both the size and number of executors. We hypothesise that a good approximation for $t_{serial}$ depends on the number of links between the nodes. This is the same as the number of edges in a fully connected graph:

$$nlinks = \frac{nexec(nexec - 1)}{2} \tag{12}$$

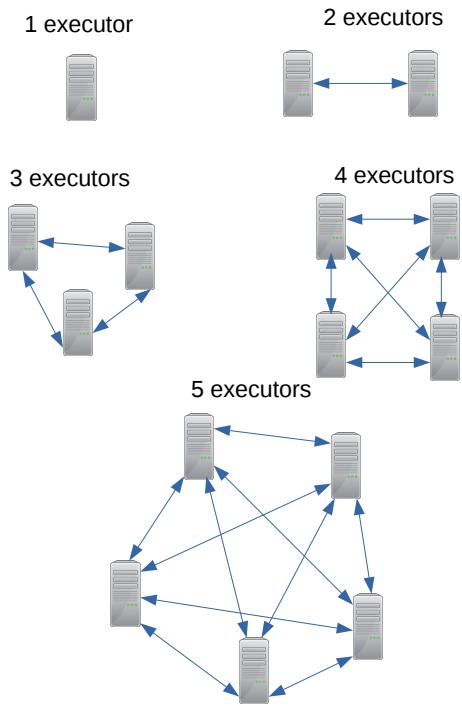

Number of boundaries: e(e-1)/2

**Figure 5.** Communication model based on fully connected graphs.

Furthermore, the serial portion becomes:

$$t_{serial} = g(Size)\left(\frac{nexec(nexec - 1)}{2}\right) \tag{13}$$

Equation (11) can be rewritten as:

$$runtime = \frac{f(Size)}{nexec} + g(Size)\left(\frac{nexec(nexec - 1)}{2}\right) \tag{14}$$

In the parallelisable part of the runtime, the function $f(Size)$ can be simplified to the complexity of the algorithm implemented for the workload. However, for the serial part, the $g(Size)$ function is unknown. The data can fit well considering that $g(Size) = b\,Size^c$, where c is a constant exponent less than 1. For better fitting, we added another coefficient, $d$, representing a constant term for a given dataset (similarly to [31]).

For linear algorithms, $f(Size) = a\,Size$, so we can rewrite Equation (14) as:

$$runtime = \frac{a\,Size}{nexec} + b\,Size^c\left(\frac{nexec(nexec - 1)}{2}\right) + d \tag{15}$$

If the workload is quadratic, $f(Size) = a\,Size^2$, and Equation (14) can be rewritten as:

$$runtime = \frac{a\,Size^2}{nexec} + b\,Size^c \left( \frac{nexec(nexec - 1)}{2} \right) + d \tag{16}$$

## 6. Experiments

### 6.1. Experimental Setup

The experimental big data cluster used in this work was designed and developed by a group of academics at Massey University, Auckland campus [38]. The hardware for this experimental big data cluster is similar to a Beowful cluster. The cluster runs on dedicated network infrastructure with dedicated switches. All other network machines are kept away from this infrastructure to reduce the network latency and unwanted network resource utilisation. The cluster was designed and developed with one master node and nine slave/worker nodes. The Hadoop cluster server and node configuration is presented in Table 2. The schematic diagram of the cluster is presented in Figure 6.

**Table 2.** Experimental configuration of the Hadoop cluster.

| Server Configuration | |
| --- | --- |
| Processor | 2.9 GHz |
| Main memory | 64 GB |
| Storage | 10 TB |
| **Node Configuration** | |
| CPU | Intel (R) Xeon (R) CPU E3-1231 v3@3.40 GHz |
| Main memory | 32 GB |
| Number of Nodes | 9 |
| Storage | 6 TB each, 54 TB total |
| CPU cores | 8 each, 72 total |
| **Software** | |
| Operating System | Ubuntu 16.04.2 (GNU/Linux 4.13.0-37-generic x86 64) |
| Hadoop | 2.4.0 |
| Spark | 2.1.0 |
| JDK | 1.7.0 |

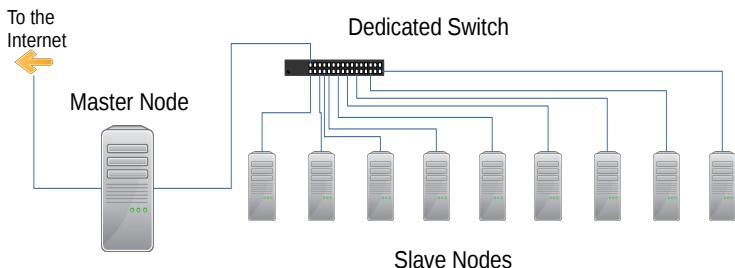

**Figure 6.** Schematic diagram of the Hadoop cluster used in the experiment.

### 6.2. Experiment Performance Evaluation

HiBench [39] is a popular big data benchmark suite that helps researchers and professionals to evaluate big data frameworks' performances. HiBench offers various characteristics and evaluates cluster deployment through comprehensive benchmarking [40]. It consists of various Hadoop programs, namely, synthetic micro-benchmarking and real-world applications. This experiment used five workloads from four different benchmark categories: Micro-Benchmark (WordCount), Machine Learning (Kmeans and SVM), Web Search (PageRank), and Graph (NWeight). The statistics of the experimental workloads

are presented in Table 3 and the workload of Spark HiBench's characteristics are presented in Table 4. Our target was to predict the Spark execution time considering the above workloads, and show how the execution time will be fitted with the proposed models. The individual workload DAG of stages and their execution are presented in the following section.

**WordCount (WC):** The WC workload performs the operation based on the Map function, which transforms the data into various representations. In HiBench, the WC input data are produced based on *RandomTextWriter*, which is contained in the Hadoop distribution. It counts the occurrences of separate words from the text or sequence file. An example of a job execution plan and its DAG of stage is presented in Figure 7. As shown in the figure, WC performed the operation in two stages; five tasks were involved in this operation.

**Table 3.** Spark HiBenchmark workload considered for this study.

| Benchmark Categories | Application | Input Data Size | | Input Samples |
|---|---|---|---|---|
| | | Multiple-Exec. | Single-Exec. | |
| Micro Benchmark | WordCount | 313 MB, 940 MB, 5.9 GB, 8.8 GB, and 19.2 GB | 3 GB, 5 GB, 7 GB, 10 GB, 12.8 GB, 14.4 GB, 16 GB, 18 GB, and 21.6 GB | - |
| Machine Learning | Kmeans | 19 GB, 56 GB, 94 GB, 130 GB, and 168 GB | 1 GB, 38 GB, 75 GB, 113 GB, 149 GB, and 187 GB | 10, 30, 50, 70, and 90 (million samples) |
| | SVM | 34 MB, 60 MB, 1.2 GB, 1.8 GB and 2 GB | 200 MB, 400 MB, 600 MB, 800 MB, 1.35 GB, 2 GB, 2.3 GB, and 2.5 GB | 2100, 2600, 3600, 4100, and 5100 (samples) |
| Web Search | PageRank | 507 MB, 1.6 GB, 2.8 GB, 4 GB, and 5 GB | 100 MB, 250 MB, 750 MB, 6 GB, 7 GB, 8 GB, 9 GB, and 10 GB | 1, 3, 5, 7, and 9 (million of pages) |
| Graph | NWeight | 37 MB, 70 MB, 129 MB, 155 MB, and 211 MB | 20 MB, 55 MB, 99 MB, 141 MB, 175 MB, 214 MB, 247 MB, 262 MB, and 286 MB | 1, 2, 4, 5, and 7 (million of edges) |

**Kmeans**: K-means is a well-known clustering algorithm that is commonly used for knowledge discovery and data mining. The Kmeans input data are a group of samples, generated by *GenKMeansDataset*, which is based on uniform and Gaussian distribution. We used various amounts of input data, such as tiny, small, and large, with the dimensions of 3 and 20; "no of cluster", 5; "max-iteration", 5; centroid, 10; and converged, 0.5. The job was executed in 19 different stages, and a sample of job DAG stages is shown in Figure 8.

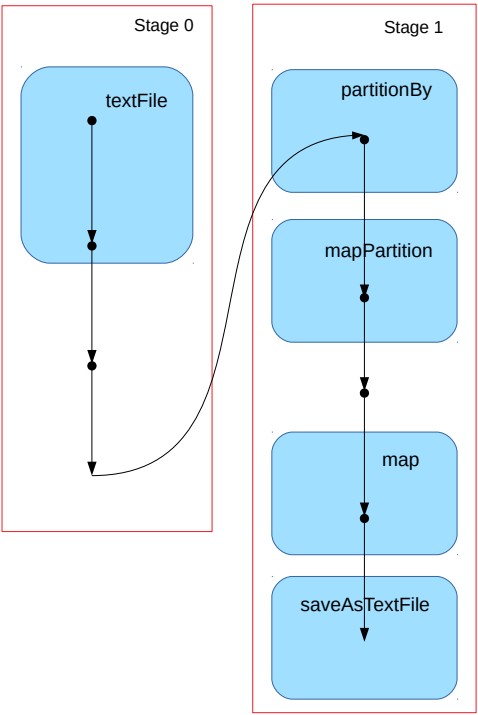

**Figure 7.** Spark stages DAG of WC.

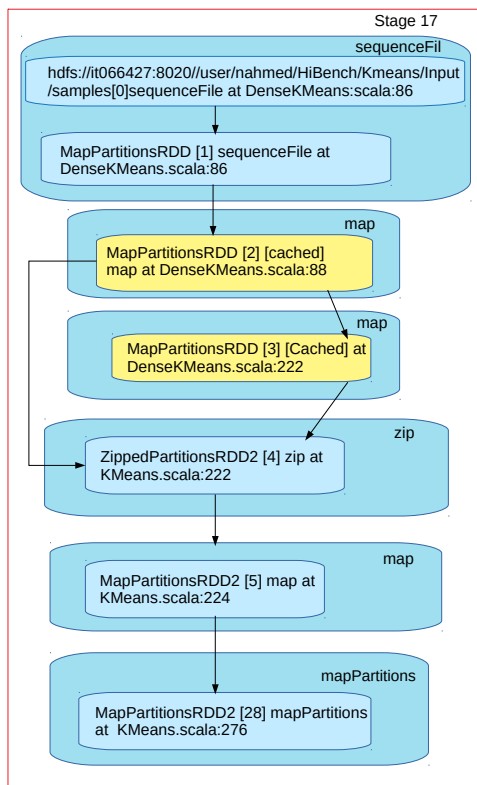

**Figure 8.** Spark stages DAG of Kmeans.

**SVM**: Support vector machines (SVMs) are used for large-scale data classification tasks. It is considered one of the standard methods of big data classification. In Spark, MLlib is used for SVM workload implementation. Its input data are generated by the SVM

*DataGenerator*. We selected the SVM parameters such as number of iterations, stepSize, and regParam and modified their values to 100, 1.0, and 0.01. The system required 213 stages to complete the task. Figure 9 shows a sample DAG of SVM.

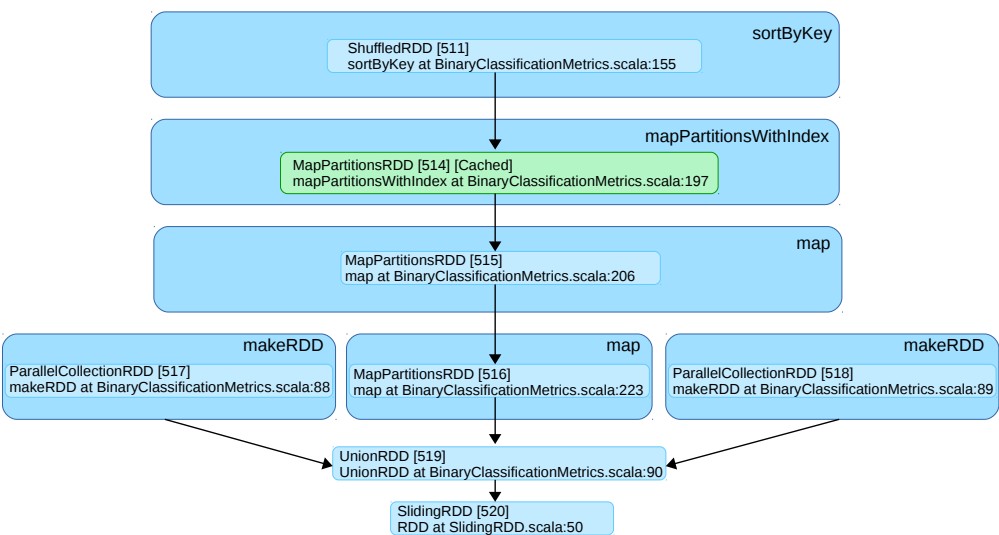

**Figure 9.** Spark stages DAG of SVM.

**NWeight**: NWeight is implemented by Spark GraphX library and pregel, and it works as an iterative parallel algorithm. It enhances the Spark RDD with a directed multigraph, which consists of properties enclosed with vertices and edges. The input files consist of millions of edges. It required eight different stages to complete the task. The workload DAG of stages is shown in Figure 10.

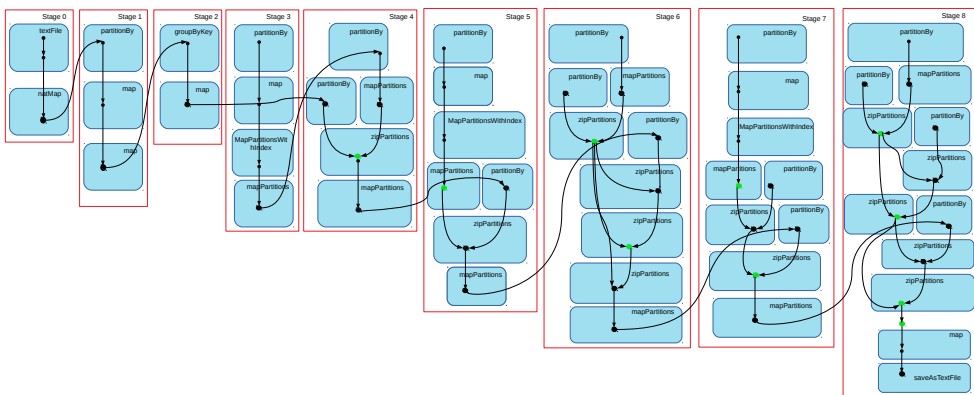

**Figure 10.** Spark stages DAG of NWeight.

**PageRank**: PageRank is a well-known page search algorithm where every page has a unique number, and an individual page is ranked as per the vote. The vote is counted when the pages are connected with the other pages. Generally, when a page is linked with several different pages, it is considered as a higher PageRank. In PageRank, the data source is generated from Web data. The hyperlinks of those data follow the Zipfian distribution. Various sets of input samples (from thousands to millions) were used in this experiment. The job was executed in four different stages. Figure 11 shows the workload DAG of stages.

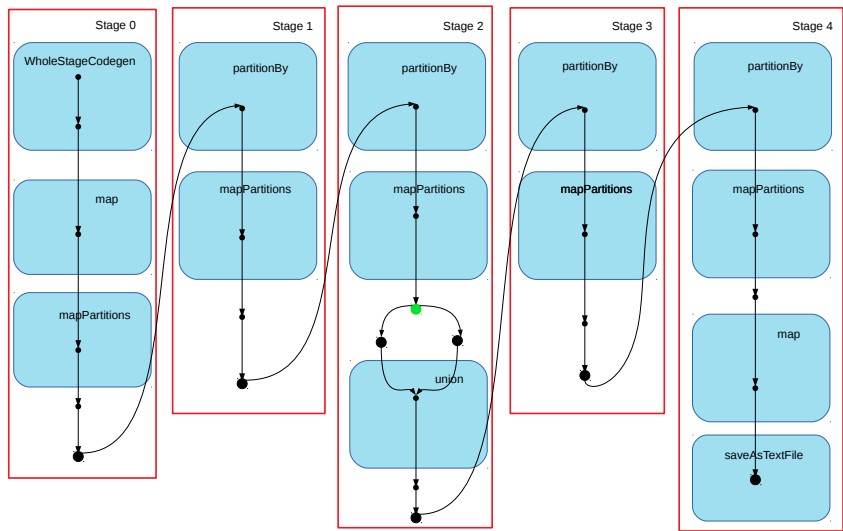

**Figure 11.** Spark stages DAG of PageRank.

**Table 4.** Workload application characteristics.

| Workloads | Stages | Parallel Stages | Collect | Serialization | Deserialization | Shuffle | Aggregate |
|---|---|---|---|---|---|---|---|
| WC | 2 | no | yes | - | - | yes | - |
| SVM | 209 | no | yes | no | yes | yes | yes |
| NWeight | 9 | yes | - | no | yes | yes | - |
| Kmeans | 20 | no | yes | yes | yes | yes | - |
| PageRank | 5 | no | - | no | yes | yes | - |

*6.3. Configuration of Parameters*

One of the challenging jobs of Spark cluster deployment is the parameter selection. There are more than 150 configurable parameters [8] in the Spark system, and each parameter plays a vital role in improving system performance. Spark's cluster performance relies on hardware infrastructure and accurate parameter selection. Performance improvements can be achieved by tuning the values of the parameters. The configuration of these parameters needs to be investigated according to the applications, amount of data, and cluster architecture. However, some influential parameters—*executors*, *executors core*, *executor memory*, *driver memory*, etc.—massively influence the system's performance. Besides, normally the number of parameters used is the default value. We have performed extensive experiments, selected the impactful parameters, and tuned their crucial factors to validate our cluster to get optimum performance. In the recent past, some studies [35,41] were carried out which illustrate the impacts and importance of the parameters. The chosen parameters for our study are listed in Table 5.

The default column in Table 5 presents the system's default configuration, the range column presents the tuned values used in this experiment, and the description column presents parameter information. There were two reasons to choose these parameters: firstly, Spark's runtime performance heavily depends on these parameters; secondly, these parameters control pivotal resources: CPU, disk read and write, and memory [42].

**Table 5.** Spark HiBenchmark parameters considered in this study.

| Parameters | Default | Range | Description |
|---|---|---|---|
| Spark.executor.memory | 1 | 12 | Amount of memory to use per executor process, in GB. |
| Spark.executor.cores | 1 | 2–14 | The number of cores to use on each executor. |
| Spark.driver.memory | 1 | 4 | Amount of memory to use for the driver process, in GB. |
| Spark.driver.cores | 1 | 3 | The Number of cores to use for the driver process. |
| Spark.shuffle.file.buffer | 32 | 48 | Size of the in-memory buffer for each shuffle file output stream, in KB. |
| Spark.reducer.maxSizeInFlight | 48 | 96 | Maximum size of map outputs to fetch simultaneously from each reduce task, in MB. |
| Spark.memory.fraction | 0.6 | 0.1–0.4 | Fraction of heap space used for execution and storage. |
| Spark.memory.storageFraction | 0.5 | 0.1-0.4 | Amount of storage memory immune to eviction expressed as a fraction of the size of the region. |
| Spark.task.maxFailures | 4 | 5 | Number of failures of any particular the task before giving up on the job. |
| Spark.speculation | False | True/ False | If set to "true" performs speculative execution of tasks. |
| Spark.rpc.message.maxSize | 128 | 256 | Maximum message size to allow in "control plane" communication, in MB. |
| Spark.io.compression.codec | snappy | lz4/lzf/snappy | Compress map output files. |
| Spark.io.compression.snappy.blockSize | 32 | 32–128 | Block size in Snappy compression, in KB |

## 7. Results and Analysis

In this part, we present the experimental findings. We have considered various amounts of data and systematically increased the number of executors to study the system's behaviour. For the results' reproducibility, each experiment was repeated at least three times, and the average execution time was taken into consideration in the final graph. We have collected the log files from the history server and calculated the job execution time using a Python script. We found that there is some fraction of the time difference between Ambari and our Python script. We have considered the most realistic time.

### 7.1. Procedure to Fit Equations

The procedure for the experiments for the HiBench workloads used for this work is summarised as follows. Firstly, for a certain workload, we ran a few jobs with different sizes using only one executor (Section 6.2). We then estimated the function $f(Size)$, which reflects the time complexity of the implemented algorithm (Section 7.2).

We ran more jobs with similar sizes as above, while varying the number of executors (Section 6.2). For each workload, we used up to 14 executors and five different sizes, chosen appropriately for each workload (Table 3).

Using the multi-parameter fitting function available in Gnuplot [43], each dataset was fitted to the following Equations: (8) (linear) or (9) (quadratic), (15) (linear) or (16) (quadratic), Amdahl (3), Gustafson (5), and Ernest [31] (Section 7.3).

The best fit for each equation above was chosen considering Rsquared and $\frac{RSE}{\mu}$ as a criterion, as discussed in Section 7.4.

### 7.2. Finding the Approximate Algorithm Complexity (f(Size))

Regarding the nominal time complexity of the algorithms used in HiBench for these experiments, some are linear and some are quadratic. SVM is typically $O(N^2)$ or even $O(N^3)$ [44]. K-means is usually quadratic [45]. The PageRank algorithm can be $O(n * m)$, where $n$ is the number of nodes and $m$ is the number of arcs [46]. WordCount is usually linear $O(N)$ [47]. The Graph (NWeight) algorithm can be either linear or quadratic, depending on the graph representation. Using an edge list, it is quadratic $O(N^2)$ [48].

In order to find the function $f(Size)$ for Equation (11), we ran several jobs using a single executor. The results are shown in Figure 12.

Based on the residual standard error (given in Gnuplot [43] as the *rms* value) of the fitting to linear or quadratic trends, it was found that only SVM and NWeight had quadratic trends. The other methods produced linear trends. The appropriate equations were fitted to the complete data. Dataset size and number of executors were used as parameters for the model.

The possible complexity for the workloads and the actual data fittings for single executors are summarised in Table 6.

**Table 6.** Time complexity for the workloads.

| Workload | Theoretical Time Complexity | Single Executor Best Fit $f(Size)$ |
|----------|------------------------------|-------------------------------------|
| WordCount | $O(N)$ [47] | linear |
| SVM | $O(N^2)$ [44] | quadratic |
| PageRank | $O(n * m)$ [46] | linear |
| Kmeans | $O(N^2)$ or $O(N)$ [45] | linear |
| NWeight | $O(N^2)$ or $O(N)$ [48] | quadratic |

### 7.3. The Full Model Fitting

After running several jobs with different HiBench workloads, we collected runtime data for various sizes and numbers of executors. The data were fitted using equations for the fully connected model: Equation (14) (replaced by Equation (15) for linear complexity or (16) for quadratic complexity); and for the special case where $c = 1$, Amdahl's Equation (3), Gustafson's Equation (5), Ernest's equation [31], and the 2D plate model using Equation (7) (replaced by Equation (8) for linear complexity or (9) for quadratic complexity). The best fit is shown in the figures below, considering the fitting criteria described in Section 7.4. Table 7 presents the Rsquared results for each equation, and Table 8 presents the RRSE results for each equation.

Figure 13 shows the graph presentation of the WordCount workload for the amounts of data between 0.3 and 19 GB. The best fit used Equation (8) and there was a draw with Amdahl's equation, yielding an Rsquared of 0.997 and an RRSE of 0.074.

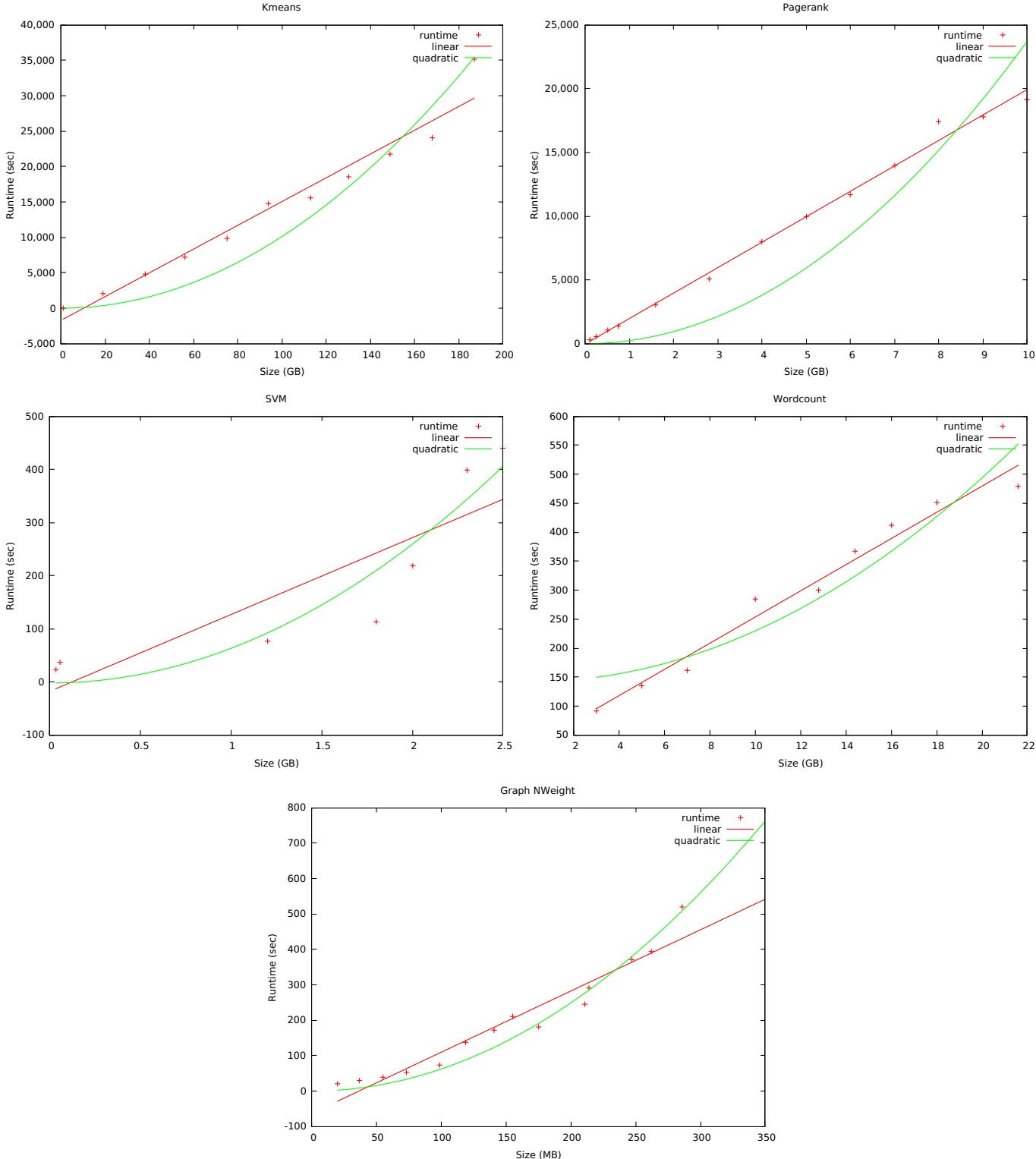

**Figure 12.** Single executor runtime complexity with different sizes.

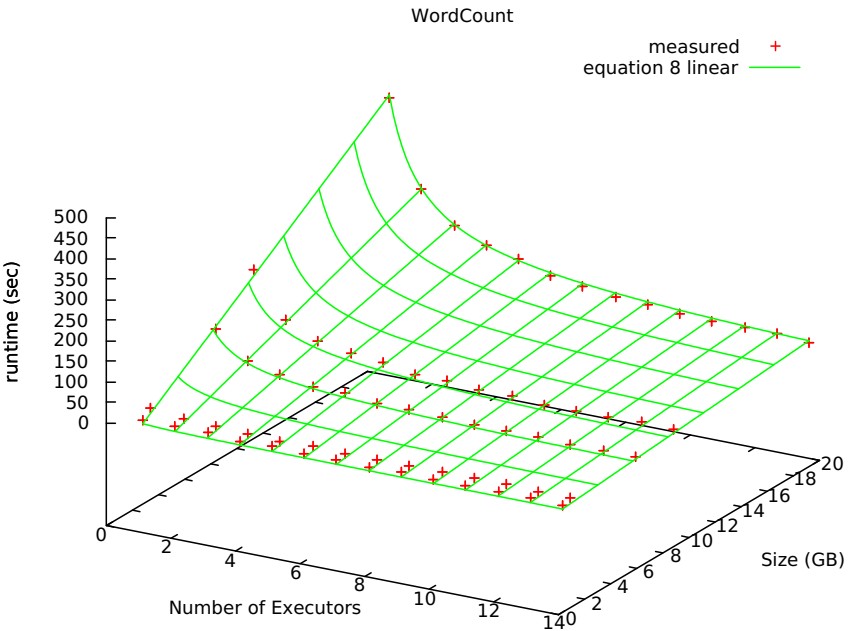

**Figure 13.** Fitting the model to WordCount workload for amount of data.

Figure 14 shows the SVM workload for sizes between 0.034 and 2 GB. Equations (9) and (16) were the best fit for the data, with an Rsquared of 0.917 and an RRSE of 0.271. The relatively high RRSE indicates that the data may be dependent on other factors, which may be investigated in future works.

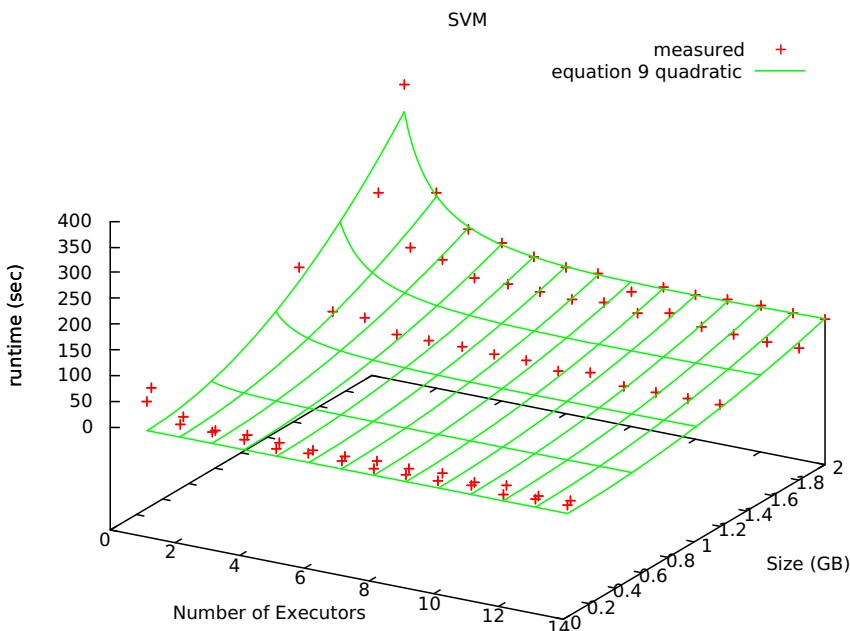

**Figure 14.** Fitting the model to SVM workload with different dataset sizes.

Figure 15 shows the PageRank workload for sizes between 0.057 and 5 GB. Equation (8) and Amdahl's were the best fit for the data, with an Rsquared of 0.990 and an RRSE of 0.113.

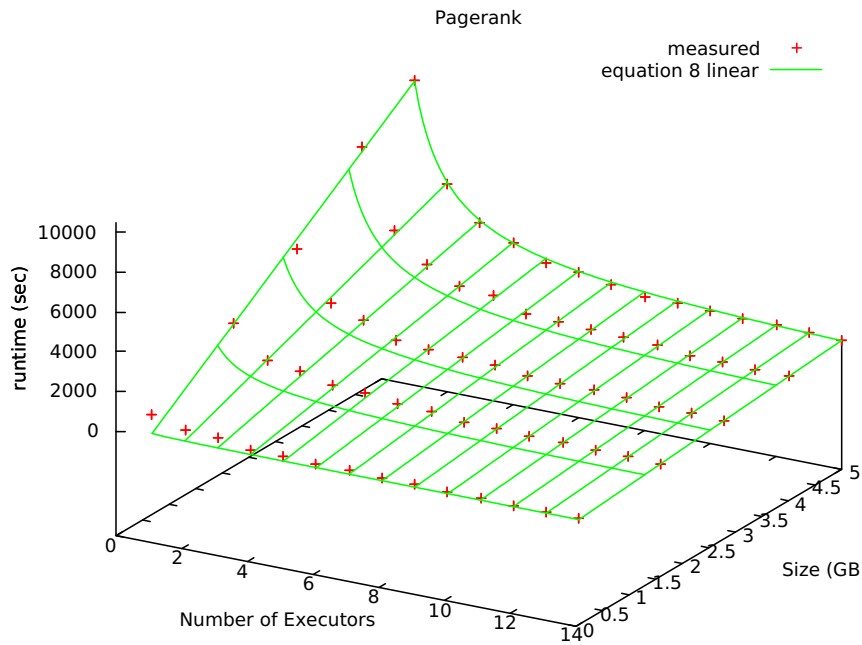

**Figure 15.** Fitting the model to PageRank workload for amount of data.

Figure 16 shows the Kmeans workload for sizes between 19 and 168 GB. Equation (8) was the best fit for the data, with an Rsquared of 0.993 and an RRSE of 0.130.

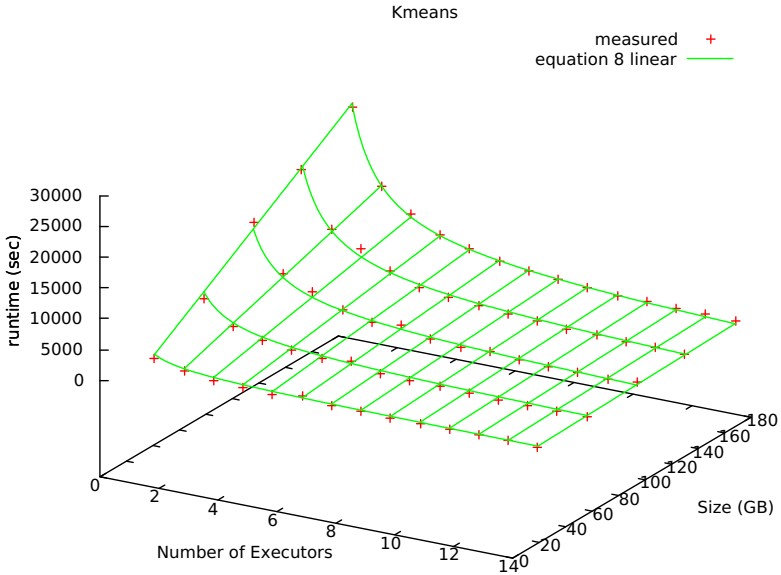

**Figure 16.** Fitting the model to Kmeans workloads with different sizes.

Figure 17 shows the Graph (NWeight) workload for sizes between 37 and 211 MB. Equation (9) was the best fit for the data with an Rsquared of 0.966 and an RRSE of 0.189.

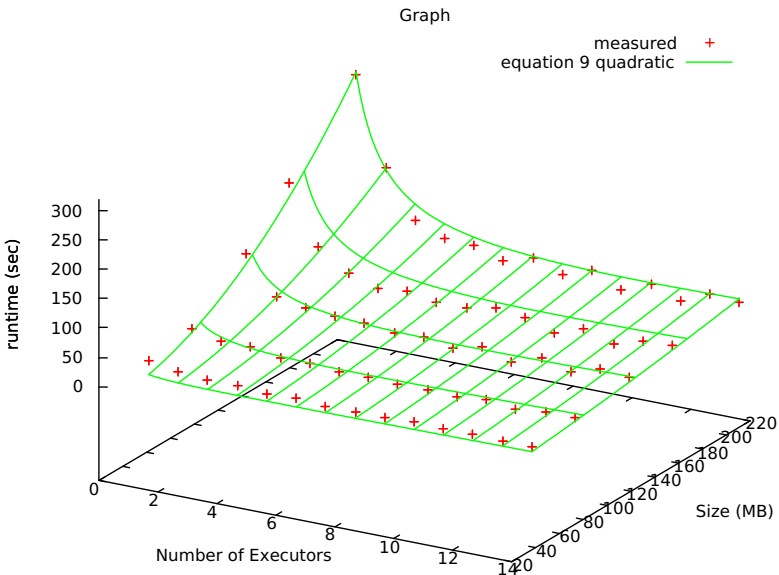

**Figure 17.** Fitting the model to Graph (NWeight) workloads of different sizes.

### 7.4. Evaluation of the Fitting Errors

The proposed models' fitting results are shown in Section 7. The nominal time complexities of different implemented algorithms in the HiBench workloads are also presented. This section illustrates the accuracy error of the proposed models, and shows the comparison results among the two proposed models and Amdahl's and Gustafson's laws. In addition, the proposed models offer an improvement over those of Ernest [31]. Our results revealed that accuracy and effectiveness of our proposed models are better than those of the Ernest models. We used both the Rsquared ($R^2$) and the relative residual standard error $\frac{RMS}{\mu}$ as metrics for the quality of the fitting. Rsquared values (also known as coefficient of determination) are calculated by the following equation:

$$R^2 = 1 - \frac{SS_{res}}{SS_{tot}} \tag{17}$$

where $SS_{res}$ is the sum of the squares of the residuals and $SS_{tot}$ is the sum of the squares relative to the mean of the data. For a perfect fitting, $SS_{res} = 0$ and $R^2 = 1$, so the closer $R^2$ is to one, the better the fitting.

The residual standard error [49] is:

$$RSE = \sqrt{\frac{\sum_{i=1}^{n}(y_i - \bar{y}_i)^2}{df}} \tag{18}$$

where $(y_i - \bar{y}_i)$ is the difference between the observed data and the predicted value using the model, and $df$ is the degrees of freedom given by the number of samples minus the number of parameters being fitted.

The relative residual standard error is:

$$RRSE = \frac{RSE}{\mu} \tag{19}$$

The *RRSE* gives a metric for the error or distance between the observed points and the ones generated by the model. The smaller the *RRSE*, the better the fit accuracy.

**Table 7.** Rsquared (Equation (17)) values for different models and workloads.

| Workload | Best Fit | Equations (15) or (16) | Equations (15) or (16) (c=1) | Amdhal Equation (3) | Gustafson Equation (5) | Equations (8) or (9) | Ernest [31] |
|---|---|---|---|---|---|---|---|
| Wordcount | linear | 0.996 | 0.996 | **0.997** | 0.996 | **0.997** | 0.995 |
| SVM | quadrat. | **0.917** | 0.912 | 0.906 | 0.887 | **0.917** | 0.847 |
| PageRank | linear | **0.990** | 0.989 | **0.990** | 0.989 | **0.990** | 0.988 |
| Kmeans | linear | 0.992 | 0.992 | 0.992 | **0.993** | **0.993** | 0.992 |
| NWeight | quadrat. | 0.964 | 0.964 | 0.956 | 0.965 | **0.966** | 0.950 |

**Table 8.** Relative residual standard error (RRSE Equation (19)) values for different models and workloads.

| Workload | Best Fit | Equations (15) or (16) | Equations (15) or (16) (c=1) | Amdhal Equation (3) | Gustafson Equation (5) | Equations (8) or (9) | Ernest [31] |
|---|---|---|---|---|---|---|---|
| Wordcount | linear | 0.083 | 0.083 | **0.074** | 0.082 | **0.074** | 0.091 |
| SVM | quadrat. | **0.271** | 0.276 | 0.285 | 0.313 | **0.271** | 0.367 |
| PageRank | linear | 0.116 | 0.118 | **0.113** | 0.121 | **0.113** | 0.127 |
| Kmeans | linear | 0.138 | 0.137 | 0.139 | 0.131 | **0.130** | 0.137 |
| NWeight | quadrat. | 0.193 | 0.193 | 0.212 | 0.190 | **0.189** | 0.226 |

*7.5. Benefits of the Proposed Models*

Our work considered three well-established equations, namely, those of Amdhal [36], Gustafson [37], and Ernest [31], as comparative models. We limited our analysis to these models while recognising the existence of alternative models in the published literature, which we deemed out of scope for the purposes of this study. As shown in Tables 7 and 8, for every workload, both proposed models (Equations (8) or (9), and (15) or (16)) had better fitting results than Ernest. Only in two cases did Amdahl's model Rsquared tie with our models, Wordcount and PageRank. Only in the case of Kmeans workload did Gustafson's model tie with one of the proposed models (Equation (8)). The results show that the two proposed models either tied with or performed better than the previously published models.

Considering the above results, the proposed models can be used as very effective tools for performance prediction, as they can offer several benefits for Spark job run time prediction using the Hadoop cluster. Several key benefits differentiate the proposed models from existing approaches. One of the crucial benefits is that using one of the proposed equations, it is possible to estimate the runtime. This can be achieved with a small number of experiments given the amount of data for the job and the chosen number of executors. The major advantage of the proposed models is that they do not require any trial-and-error approach, nor do they require large amounts of training or test data that are usually needed by machine learning models. Both models can capture the performance characteristics of a large number of complex workloads and are capable of predicting the runtime with good accuracy. Finally, the results also show that the models are highly effective, generic, and platform agnostic. Based on these models, it is possible for the managerial teams of big-data-driven organisations to minimise the time of their systems' configuring processes, plan and schedule large jobs by allocating critical resources for the clusters, and choose appropriate numbers of executors to maximise resource utilisation.

## 8. Conclusions

This paper proposed and investigated parallelisation models with enhanced capabilities for predicting the runtime performance of Apache Spark for several workloads running on Hadoop clusters. The configuration of Spark parameters is a complex and challenging task for the users. The system's performance mainly depends on the user's choice and targets.

To overcome this challenge, we proposed two models based on a function of the two most important parameters, the number of executors and the size of the job. A significant contribution of this work is the finding that with limited data points, one can fit the data into simple equations and understand the pattern of communication between the nodes when

running Spark jobs in a Hadoop cluster. We have found that the communication patterns can vary wildly between different workloads. This is expected, as different algorithms have different requirements from the Hadoop cluster. However, it can be noted that all the workloads used in the experiments could fit one of the two proposed models. The experimental results show that all the workloads could be fitted very accurately using the models proposed and completely outperformed the Ernest model. For two of the workloads (SVM and NWeight), the Rsquared values produced were lower than those the other three, alongside relatively high residual standard error. The two models fit the data better or at least as well as other alternative models (Amdahl, Gustafson, and Ernest).

However, the proposed models should be evaluated with other benchmark workloads, such as SQL and streaming. Due to time constraints, we considered only five workloads and selected a limited number of suitable Spark parameters. As future work, we have a plan to test the proposed model on the latest version of Apache Spark. Besides, we aim to add more suitable Spark parameters and workloads, and compare the proposed models with machine learning models. Furthermore, we intend to expand the experiments to other HiBench workloads to determine which equations are more suitable for which workloads.

**Author Contributions:** Conceptualization: N.A. and A.L.C.B.; methodology: N.A. and A.L.C.B.; resources: N.A.; validation: N.A. and A.L.C.B.; formal analysis: N.A. and A.L.C.B.; investigation: N.A. and A.L.C.B.; Data curation: N.A. and A.L.C.B.; writing—original draft preparation: N.A. and A.L.C.B.; writing—review and editing: N.A., A.L.C.B., M.A.R. and T.S.; visualization: N.A. and A.L.C.B.; supervision: A.L.C.B., M.A.R. and T.S. All authors have read and agreed to the published version of the manuscript.

**Funding:** This work was supported by the REaDI funding [Project code: 96670].

**Institutional Review Board Statement:** Not applicable.

**Informed Consent Statement:** Not applicable.

**Data Availability Statement:** Data are contained within the article. However, the correspondence author can be contacted for more details.

**Acknowledgments:** This work was supported in part by the Massey University Doctoral Scholarship.

**Conflicts of Interest:** The authors declare no conflict of interest.

## Abbreviations

The following abbreviations are used in this manuscript:

| | |
|---|---|
| SVM | Support vector machines |
| API | Application programming interface |
| SQL | Structured query language |
| HDFS | Hadoop distributed file system |
| RDD | Resilient distributed datasets |
| MLlib | Machine learning library |
| CPU | Central processing unit |
| I/O | input/output |
| UC | University of california |
| AMP | Algorithms, machines and people |
| DAG | Directed acyclic graph |
| YARN | Yet another resource negotiator |
| PERIDOT | Performance predIction moDel fOr Spark applicaTions |
| NEXEC | Number of executor |
| SQRT | Square root |
| 2D | Two dimensional |
| GHz | Gigahertz |
| TB | Terabyte |
| RAM | Random access memory |

| DDR | Double data rate |
| --- | --- |
| GB | Gigabyte |
| MB | Megabyte |
| WC | WordCount |
| Exec | Executor |
| MOP | Multi-object optimization |

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
