# Peer review of "An Enhanced Parallelisation Model for Performance Prediction of Apache Spark on a Multinode Hadoop Cluster"

_2504-2289, doi:10.3390/bdcc5040065_

Round 1

Reviewer 1 Report

The authors added a comparison to Ernest and show that their approach works (a bit) better than Ernest. However, they did not include a explanation why other systems where disregarded in the evaluation.

Although many typos and grammar mistakes have been fixed, there are still numerous spelling errors present.

Author Response

Comments and Suggestions for Authors

The authors added a comparison to Ernest and show that their approach works (a bit) better than Ernest. However, they did not include a explanation why other systems where disregarded in the evaluation.

Answer: In response to the above, the authors have added the following statement in section 7.5. (“One of the major advantages of this model is that it does not require any trial-and-error approach. It can also be noted from the results that the models do not require large amounts of training or test data needed by machine learning models.”)

The authors found that the proposed models are similar to Ernest technique and their comparative results are added already in the manuscript. As per our best knowledge, all other published models are out of the scope for comparison. The work in the manuscript proposes a simple equation fitting to experimental data, and therefore we compared to the three equations published in the literature, namely Amdhal, Gustafson, and Ernest equations. However, the authors have the plan to do more investigation on this work with the comparison of machine learning models in the future.  

Although many typos and grammar mistakes have been fixed, there are still numerous spelling errors present.

Answer: The authors have updated the manuscript and corrected all the typos and grammatical mistakes we could find.

Reviewer 2 Report

Authors have addressed majority of the comments given by the reviewer. Though, I'm still not convinced for the contributions and novelty of the article.

Author Response

Comments and Suggestions for Authors

Authors have addressed majority of the comments given by the reviewer. Though, I'm still not convinced for the contributions and novelty of the article.

Answer: The authors appreciate the reviewer’s feedback. To address the reviewer's comment on the contribution and novelty aspect, the authors would like to emphasise that we have developed two new models and have done extensive experimental work.

This has been in addition to an extensive statistical analysis to demonstrate that this model would generalise well with numerous domains in the IT industry. The authors also believe that the contribution from this paper is novel. We acknowledge similar models were proposed by Amdahl, Gustafson, and Ernest. However, we have clearly demonstrated an improvement in accuracy over these models.

Reviewer 3 Report

Dear Authors,

Please see below my remarks about your article "Enhanced Parallelization Model for Performance Prediction of Apache Spark on a Multinode Hadoop Cluster".

1. In the Introduction section you should present to the readers the general view of the Big Data market, so that they understand the importance of Spark in the big data engines ecosystem.
I recommend you to add a new distinct paragraph where you present the Spark market share by comparing to the other similar tools (competitors for Spark).
As I know, Apache Spark has less than 10% of the entire market, so you have to justify your choice for this article by citing official reports.

2. Sections "7.5. Benefits of the Proposed Models" and "8. Conclusion" must be improved. From my point of view, at this moment, these two sections are the weak part of the article.
Thus, I recommend you the following actions:
- include, as a limitation, the fact that you tested your model only on one version of Spark.
- include, as future directions:
1) comparative testing on different versions of Spark;
and
2) useful information for heterogenous IT integration (cite here https://doi.org/10.12948/issn14531305/23.4.2019.02, https://doi.org/10.1109/JPROC.2020.3035874, http://clausiuspress.com/conferences/LNEMSS/EMELS%202020/EMELS008.pdf). By citing these references, you "open" your research results to the future research directions and this is a very important aspect for a scientific article.

Kind Regards!

Author Response

Comments and Suggestions for Authors

Dear Authors,

Please see below my remarks about your article "Enhanced Parallelization Model for Performance Prediction of Apache Spark on a Multinode Hadoop Cluster".

In the Introduction section you should present to the readers the general view of the Big Data market, so that they understand the importance of Spark in the big data engines ecosystem.
I recommend you to add a new distinct paragraph where you present the Spark market share by comparing to the other similar tools (competitors for Spark).
As I know, Apache Spark has less than 10% of the entire market, so you have to justify your choice for this article by citing official reports.

Answer:  The authors acknowledge that stating the market share of Apache Spark could be an interesting and useful fact to include in the paper. The authors added the Spark market share information in section 2. The added information is in blue text. (“Spark opens a new window to process data faster and its uses are in data analytics, big data processing, and Machine Learning. The major advantage of Apache Spark for machine learning is its end-to-end capabilities. As per the Datanyze market research, Apache Spark market share is almost 6.40% with more than 2770 companies globally. As per enlyft data, 59% of customers of Apache Spark are in United State, 6% in the United Kingdom, and 6% in India.”). The information comes from the following sources:

Datanyze- (https://www.datanyze.com/market-share/big-data-processing--204/apache-spark-market-share)

enlyft data-  (https://enlyft.com/tech/products/apache-spark). 

The authors have felt somewhat hesitant to add this information not the paper since the models developed in this paper are generic and platform agnostic. We were concerned that emphasising Apache Spark might give the readers the impression that our work applies onto to Apache Spark, when in fact our models are applicable to any cluster or parallel machine technology. Apache Spark was used only due to the fact that the authors have access to a physical cluster running Spark.

  1. Sections "7.5. Benefits of the Proposed Models" and "8. Conclusion" must be improved.

 From my point of view, at this moment, these two sections are the weak part of the article. Thus, I recommend you the following actions:
- include, as a limitation, the fact that you tested your model only on one version of Spark.
- include, as future directions:
1) comparative testing on different versions of Spark;
and

Answer: The authors appreciate the observation of the reviewer. The authors have added an explanation of the limitations of this work as well as the future direction in the Conclusion section.

2) useful information for heterogeneous IT integration (cite here

Answer: The authors appreciate the suggested citations by the reviewer. We have carefully examined the suggested papers for inclusion. However, we have found that the suggested papers do not quite match the scope of our study. We provide a brief summary below of why we think that these papers do not quite line up with the goals of our study:

https://doi.org/10.12948/issn14531305/23.4.2019.02,
Summary: This paper discusses the physical integration of heterogeneous web-based data. Crucially, the paper and does not present any aspects regarding performance or modeling of the parallelisation for such systems. We, therefore, believe that this paper is out of scope regarding the main topic of our manuscript.

https://doi.org/10.1109/JPROC.2020.3035874,
Summary: This paper is well written and presented, but it is again out of the scope of our manuscript. This paper discusses a programming paradigm for heterogeneous systems, and does not discuss performance modeling which is the crux of our study.

Summary: This paper is a broad study about heterogeneous data integration. The paper does not delve into the details of the runtime performance of a big data system. As such, this paper is also not in alignment with the scope of our work.

 By citing these references, you "open" your research results to future research directions and this is a very important aspect for a scientific article.

Round 2

Reviewer 3 Report

Dear Author(s),

I think you addressed most of my constructive recommendations from the previous round of review.

I have just one minor remark regarding the chapter "7.5. Benefits of the Proposed Models". Here you should insist on presenting your findings related to the the other results from the literature. I think and a short paragraph with 4-5 rows would be useful for the readers.

Kind Regards!

Author Response

Comments and Suggestions for Authors

Dear Author(s),

I think you addressed most of my constructive recommendations from the previous round of review.

I have just one minor remark regarding the chapter "7.5. Benefits of the Proposed Models". Here you should insist on presenting your findings related to the the other results from the literature. I think and a short paragraph with 4-5 rows would be useful for the readers.

Answer: In response to the above, the authors have added the first paragraph in section 7.5. The authors would like to thanks the reviewer for his valuable suggestions to make this manuscript informative, interesting, and attractive for the reader.

Our work considered three well-established equations, namely Amdhal, Gustafson, and Ernest as comparative models against our proposed models. We limited our analysis to these models while recognising the existence of alternative models in the published literature, which we deemed out of scope for the purposes of this study. As shown in tables 7 and 8, for every workload, both proposed models (equations 8 or 9, and 15 or 16) have better fitting results than Ernest. Only in two cases, Amdahl’s model Rsquared tied with our models, Wordcount and PageRank. Only in the case of Kmeans workload, Gustafson’s model is tied with one of the proposed models (equation 8). The results show that the two proposed models either tied or performed better than the previously published models.

Considering the above results, the proposed models can be used as a very effective tool for performance prediction as they can offer several benefits for the Spark job run time prediction on the Hadoop cluster. Several key benefits differentiate the proposed models from existing approaches. One of the crucial benefits is that using one of the proposed equations, it is possible to estimate the runtime. This can be achieved with a small number of experiments given the data size of the job and the chosen number of executors. The major advantage of the proposed models is that they do not require any trial-and-error approach, nor require large amounts of training or test data that are usually needed by machine learning models. Both models can capture the performance characteristic of a large number of complex workloads and are capable of predicting the runtime with good accuracy. Finally. the results also show that the models are highly effective, generic, and platform agnostic. Based on these models, it is possible for the managerial team of big-data-driven organisations to minimise the time of the system configuration,  plan and schedule large jobs by allocating critical resources for the clusters, and choose an appropriate number of executors to maximise resource utilization.

This manuscript is a resubmission of an earlier submission. The following is a list of the peer review reports and author responses from that submission.

Round 1

Reviewer 1 Report

In this paper the authors propose two models to predict the runtime of Apache Spark jobs. The models are based on the workload (the program), and the number of executors in the cluster. Starting from Amdahl's and Gustafson's laws, the authors build their models by first estimating the complexity of the given program, by fitting it to a quadratic or linear runtime model. Additionally the authors consider two communication patterns: one where a node in the cluster communicates only with a few "neighbors" and one where every nodes communicates with all others.
The proposed models are evaluated using 5 HiBench workloads.

In general the paper is well structured and easy to follow. The idea of the model is well described and comprehensively presented. 
Although the paper is easy to read, it requires some additional extensive proof reading, as it contains many errors or grammar inconsistencies (wrong capitalization, plural when singular, past vs present tense, ...). The experiments are carried out on 5 HiBench workloads, but the results achieved by the proposed models of this paper are only compared to equations proposed by Amdahl and Gustafson. The Ernest paper presented a similar model (in their Eq. 1 in the NSDI paper) that considers the number of executors and the data size. I think a comparison with other state of the art prediction models would improve the evaluation.

Some more detailed comments are:  
L29: RDDs are not an engine, they are an abstraction. Spark is the engine.
L42: what is "size" here? Should this be "workload size and number of executors"?
L95 (and others): It's "HiBench" only
L111: These are programming languages, not libraries
L119-127: What is Apache DAG ? The DAG is the graph representation of the data flow.
L140: assigns -> applies
L142: what is executor size? What are "numbers"?
L194&195: Earnest -> Ernest
L266: "Where a and b are constants" Are they really constants or rather coefficients that differ from one application to another?
L314 Eq14: I think "a" is missing in the numerator
L331: There are confusing details on the cluster: is 64GB RAM or 32? Is it 10TB storage or 60?
Table 3: Parallel stages: From the presented DAGs I see that only NWeight have parallel stages.
L390: -> "Besides, normally for the majority of parameters the default value is used" ?
Table 4: For the parameters with ranges, like spark.memory.fraction: They influence the runtime as the authors mentioned and I agree. However, they are not included the presented model. So, how to tune them and does the presented prediction model still fit them?  For spark.speculation a closing quotation mark is missing in the Description column.
L475: "It can be seen from the results that the model does not require any training or test data". In my point of view, that's not completely true. I consider all experiments in the bullets above as a training step. 
L481: "maximise these resources" -> "maximise resource utilization"?

Reviewer 2 Report

The paper proposed two distinct parallelization models for performance prediction of spark jobs based on 2D plate boundary approach. The evaluation was done on real spark cluster with synthetic benchmark hibench. 

I have several comments to the author. 

  1. I do not find any novelty or notable contributions in the manuscript. for instance, in introduction, the contributions item 2 and 3 are just experimental settings and configurations used and results observed. Authors really need to work on improving the contributions.
  2. Authors major contributions are merely less than a page (9-10). Rest is all background, literature and experimental part.
  3. Authors need to use any of existing algorithms to compare against their proposed parallelization models. As several works have been done on spark job performance and runtime prediction.
  4. It is not clear what authors wanted to predict, i.e., performance or runtime. Abstract says, runtime and job size but evaluation do not show any of such thing. Similarly, manuscript title says performance prediction.
  5. Several places in the manuscript uses "performance of the execution time" which is not even an appropriate sentence. (please see page4)
  6. Please merge section 2 and 3 and please reduce it. it makes the paper very boring to readers with un-necessary details, e.g., if you added table1 is sufficient to provide neccessary information and text before table1 can be reduced alot. 
  7. I suggest authors to clarify contributions and reduce the section3 specifically.
  8. Extensive English editing is required. I found dozens of grammar and typos in the manuscript. (please see "[19] is presented two" "which is consists of"
  9. Section 4 263-264 incorrect sentence. 
  10. Quality of the manuscript is very poor some figures resolution is way beyond publishing quality.
  11. The paper organization at the end of introduction misses the section 4.

Reviewer 3 Report

Dear Author(s),
Please find below my recommendations regarding your manuscript proposal entitled "Enhanced Parallelization Model for Performance Prediction of Apache Spark on a Multinode Hadoop Cluster".

Within section "1. Introduction" you should clearly define the following important issues:
- the research gap: please describe which is the research gap covered by your research, so that the readers are informed about the general goal of your article;
- the RQ (Research Question): please better define the research question, because you try to publish an article in a high-ranked scientific journal and the standards require to define the RQ. The RQ must be better argued, based on the previous literature. At this moment, the RQ seems to be based on your own feelings.
Thus, in the Introduction section I recommend you to improve the general context by citing the following relevant references: https://www.techscience.com/cmc/v66n3/41074 (this article is about cloud computing), https://doi.org/10.12948/issn14531305/21.1.2017.03 (this article is about software orchestration), https://doi.org/10.1016/j.jss.2021.111028 (this article is about spark performance in public clouds), https://doi.org/10.1109/MASCOTS50786.2020.9285944 (this article is about performance prediction for data-driven workflows on Apache Spark), https://doi.org/10.1007/978-3-030-59612-5_2 (this article is about a performance prediction model for Spark applications).
I consider that your Introduction section really needs the recommended references in order to improve the presentation of the context research.
Also, you should argue why you chose Apache Spark for your analysis. Please present the market share of the Spark, so that the readers understand the importance of your approach.

Section "3. Background Study" has only one sub-section: "3.1. Amdahl’s Law and Gustafson’s Law".
Normally, a section should have at least two distinct sub-sections.
Please revise and correct this aspect.

An important issue is that the readers don't understand what you want to do. The article seems to be a descriptive one, without any goal.
I recommend you to insert a new (sub)section entitled "Research methodology" and here you should shortly describe the methodology and the steps involved in your research.

Within the section "6.3 Parameters Configuration" you say at rows 395-400: "The default column in table 4 presents system default configuration, and the range column presents the tuned values used in this experiment, and the description column presents parameters information. There were two reasons to choose these parameters; firstly, Spark runtime performance heavily depends on these parameters; secondly, these parameters can primarily fulfil available resources: CPU, disk read and write, and memory."
Please add some references to justify your choice for these selected parameters.

In the Conclusion section you should present the following important issues:
- the research limitations (here you should be honest and shortly describe the limitations of your research);
- the managerial implications (here is the place where you can "sell" your findings to the readers).

Dear Author(s),
Please consider all the above remarks as being constructive recommendations in order to improve the general quality of your manuscript proposal.
Kind Regards!